# Consequences of COVID-19 on Adipose Tissue Signatures

**DOI:** 10.3390/ijms25052908

**Published:** 2024-03-02

**Authors:** Sontje Krupka, Anne Hoffmann, Mariami Jasaszwili, Arne Dietrich, Esther Guiu-Jurado, Nora Klöting, Matthias Blüher

**Affiliations:** 1Helmholtz Institute for Metabolic, Obesity and Vascular Research (HI-MAG), Helmholtz Zentrum München, University of Leipzig and University Hospital Leipzig, 04103 Leipzig, Germanymatthias.blueher@medizin.uni-leipzig.de (M.B.); 2Medical Department III—Endocrinology, Nephrology, Rheumatology, University of Leipzig Medical Center, 04103 Leipzig, Germany; 3Clinic for Visceral, Transplantation and Thorax and Vascular Surgery, University Hospital Leipzig, 04103 Leipzig, Germany

**Keywords:** COVID-19, SARS-CoV-2 infection, adipose tissue, obesity

## Abstract

Since the emergence of coronavirus disease-19 (COVID-19) in 2019, it has been crucial to investigate the causes of severe cases, particularly the higher rates of hospitalization and mortality in individuals with obesity. Previous findings suggest that adipocytes may play a role in adverse COVID-19 outcomes in people with obesity. The impact of COVID-19 vaccination and infection on adipose tissue (AT) is currently unclear. We therefore analyzed 27 paired biopsies of visceral and subcutaneous AT from donors of the Leipzig Obesity BioBank that have been categorized into three groups (1: no infection/no vaccination; 2: no infection but vaccinated; 3: infected and vaccinated) based on COVID-19 antibodies to spike (indicating vaccination) and/or nucleocapsid proteins. We provide additional insights into the impact of COVID-19 on AT biology through a comprehensive histological transcriptome and serum proteome analysis. This study demonstrates that COVID-19 infection is associated with smaller average adipocyte size. The impact of infection on gene expression was significantly more pronounced in subcutaneous than in visceral AT and mainly due to immune system-related processes. Serum proteome analysis revealed the effects of the infection on circulating adiponectin, interleukin 6 (IL-6), and carbonic anhydrase 5A (CA5A), which are all related to obesity and blood glucose abnormalities.

## 1. Introduction

Recent studies have highlighted the potential involvement of adipocytes in the infection process of severe acute respiratory syndrome coronavirus-2 (SARS-CoV-2), the virus responsible for coronavirus disease-19 (COVID-19) [1,2]. It has been observed that people with obesity and type 2 diabetes (T2D) are more susceptible to severe courses of COVID-19 and higher rates of mortality [3,4]. However, the impact of the virus on adipose tissue (AT) is still under investigation, and it remains unclear whether the increased susceptibility is directly linked to obesity or its associated comorbidities [5].

SARS-CoV-2 RNA has been recently identified in post-mortem samples of human AT [2], suggesting that adipocytes may represent a virus reservoir for COVID-19. Moreover, in an experimental hamster model, AT SARS-CoV-2 propagation was associated with specific changes in lipid metabolism and alterations in plasma lipidome [2]. Indeed, transmembrane protease serine subtype 2 TMPRSS2 and ACE2 play crucial roles in SARS-CoV-2 infection [6] and are expressed in adipocytes, making them susceptible to viral entry [7,8]. Thus, AT could act as a reservoir for the virus and contribute to prolonged or more severe disease symptoms in people with higher body fat mass. The interaction between SARS-CoV-2 and adipocytes raises concerns about the impact on metabolic health. COVID-19 may worsen metabolic disturbances commonly seen in obesity, such as insulin resistance, hypertension, and dyslipidemia [9]. These metabolic abnormalities may contribute to the increased severity of COVID-19 in individuals with obesity.

In light of recent evidence suggesting the involvement of adipocytes in the infection process of SARS-CoV-2 [1,2,10], we tested the hypothesis that COVID-19 affects AT biology. Subsequently, we asked whether COVID-19 infection may have distinct effects on visceral compared to abdominal subcutaneous fat depots. A better understanding of the potential interplay between COVID-19 infection and postulated alterations in AT may be particularly relevant to better understanding why people with obesity have a higher risk for adverse COVID-19 outcomes compared to people with normal weight. Our study aims to elucidate the consequences of COVID-19 on abdominal visceral and subcutaneous AT histology and gene expression signatures in 27 human AT donors who underwent bariatric surgery. To distinguish between the effects of the SARS-CoV-2 infection from those of the host immune response to COVID-19 vaccination, we included three groups of people with obesity that (1) did not have a history of both COVID-19 infection and vaccination, (2) had COVID-19 vaccination but no previous infection, and (3) had a COVID-19 infection and vaccination. Our study adds to the current knowledge from post-mortem AT analyses [1,2] by including living AT donors.

## 2. Results

### 2.1. Clinical and Metabolic Characteristics of Study Cohort

To test the hypothesis that a history of COVID-19 infection is associated with distinct changes in AT morphology and gene expression patterns, we took advantage of the continuously and prospectively recruiting Leipzig Obesity BioBank [11]. We were able to obtain paired visceral and subcutaneous AT biopsies from donors with different histories of COVID-19 infection and vaccination. To isolate the effects attributable to infection alone, we focused on two specific comparisons, as shown in Figure 1. We compared group 3, which included individuals who were both infected and vaccinated, with group 2, which included individuals who were vaccinated against COVID-19 but were not infected. To investigate the effects of vaccination and/or COVID-19 infection, we compared group 3 with group 1, which represented individuals who were neither infected nor vaccinated. This allowed us to evaluate the impact of vaccination when concurrent with infection.

Twenty-seven participants (15 women and 12 men) with obesity (mean BMI 44.1 ± 4.7 kg/m^2^) were characterized anthropometrically and metabolically (Table 1, Appendix A). Parameter comparisons based on normal distribution were conducted between all three groups. No significant differences were found in the phenotype parameters across the three groups.

Adipose tissue from both fat depots was tested for viral RNA using a previously reported method [2]. However, no viral RNA was detected in either SAT or VAT.

### 2.2. Smaller Average Adipocyte Size in VAT and SAT of COVID-19 Infected Patients

To investigate the impact of COVID-19 infection on morphological parameters, we measured adipocyte size using hematoxylin and eosin (HE)-stained slides from subcutaneous (SAT) and visceral adipose tissue (VAT) (Figure 2).

Group differences in adipocyte size were observed in both AT depots. The AT area analysis showed that individuals who were infected and vaccinated had a reduced average adipocyte size compared to the other groups (Figure 2). Vaccination status was not associated with adipocyte size.

### 2.3. History of COVID-19 Infection Is Associated with a Distinct AT Gene Expression Signature That Is Fat Depot-Specific

By using RNA-sequencing data, we tested the hypotheses that a history of COVID-19 affects the AT transcriptome, and that gene expression patterns are fat depot-specific. First, we compared group 3 IV with group 2 NV to show the effect of infection alone and with group 1 NN to show the effect of infection when patients were vaccinated. We identified various differentially expressed genes (see Appendix A) based on an absolute log2 fold change (FC) greater than |0.5|, while significance is indicated by an adj. *p*-value (adj.pval) less than 0.05 in AT.

The analysis reveals that infection has an impact on the number of differentially expressed genes in SAT. The number of differentially regulated genes is significantly higher when vaccination is excluded (n = 52), as shown by the comparison of group 3 IV with group 2 NV (Figure 3A). When comparing group 3 IV with group 1 NN, indicating the impact of vaccination, we detected a lower number of differentially regulated genes (n = 9), indicating a positive effect of vaccination on gene expression profile (Figure 3B; see Appendix A).

Five genes, capping protein regulator and myosin 1 linker 2 (CARMIL2), C-C motif chemokine receptor 7 (CCR7), niban apoptosis regulator 3 (NIBAN3), sterile alpha motif domain containing 3 (SAMD3), and membrane spanning 4-domain A1 (MS4A1), are altered by infection independent from vaccination because they overlap in both comparisons (Figure 3C). The most regulated gene (highest FC) is the epithelial splicing regulatory protein 1 (ESRP1) gene, which is regulated due to infection and vaccination (Figure 3B). In contrast, the eomesodermin (EOMES) gene is overexpressed, and nucleoside diphosphate kinase A (NME1), polycystic kidney disease 1-like 3 (PKD1L3), as well as nanos homolog 1 (NANOS1) are down-regulated in the infection group.

In VAT, we observe a similar trend: when comparing group 3 NN with group 2 NV (Figure 4A), the analysis reveals a lower number (n = 2) of differentially regulated genes than in comparison with group 1 NN (n = 6, Figure 4B). Both the gene *killer cell lectin-like receptor subfamily D*, *member 1* (*KLDR1*), and *myosin light chain 2* (*MYL2*) are significantly different expressed in both comparisons. However, the fold change and significance *p*-value are higher when comparing group 3 IV with group 2 NV, indicating an enhanced influence of infection on the number of regulated genes in VAT, which might be suppressed by vaccination.

### 2.4. Subcutaneous Adipose Tissue Immune System Pathways Are Affected by Vaccination

Subsequently, we investigated the impact of infection or vaccination on specific pathways by conducting gene enrichment analysis utilizing the Gene Ontology (GO) database of biological processes. Mainly, immune system pathways (Figure 5) are influenced by infection in SAT.

The pathway enrichment analysis revealed significant changes in multiple immune-related pathways, indicating a robust and intricate immune response. Key findings include heightened T-cell activation, lymphocyte activation, and broader cell and leukocyte activation, suggesting an orchestrated immune cell response. The response to biological stimuli and signal transduction pathways indicates targeted reactions to living organisms and active cellular signaling, respectively.

Due to the limited number of differently expressed genes in VAT, enrichment analysis results were not recorded.

### 2.5. Effects of COVID-19 History on Serum Protein Profile

The provided Normalized Protein Expression (NPX) values from Olink^®^ Proximal Extension Assay (PEA) panels Inflammation, Cardiometabolic, and Cardiovascular II (see Appendix A) were used to analyze the differential protein expression in serum, based on absolute log2 fold change (FC) greater than |0.5|, with significance indicated by an adj. *p*-value (adj.pval) less than 0.05 (see Appendix A). Only 2 serum protein levels out of 276 were different across groups, as shown in Figure 6. Interleukin-6 (IL-6) and carbonic anhydrase 5A (CA5A) concentrations are lower in infected patients when vaccinated (group 3 IV vs. group 1 NN) compared to controls. IL-6 was detected twice because the protein is located on two panels out of three used panels.

Adiponectin was found to be highly regulated during COVID-19 infections [1,12]. However, it was not part of the proteomics panel. Therefore, we directly measured circulating serum adiponectin levels by ELISA.

As shown in Figure 7, we found that patients with a COVID-19 infection history have the lowest adiponectin levels (group 3, IV), followed by vaccinated subjects, and the highest levels were detected in controls. Essentially, although the difference was not statistically significant, there was a trend toward decreased adiponectin levels when patients were infected.

## 3. Discussion

Patients with obesity have a greater risk of more severe courses and outcomes from SARS-CoV-2 infection. Higher AT mass may represent a mechanistic link between obesity and the adverse effects of COVID-19, particularly because SARS-CoV-2 is likely to infect AT [2,13,14,15]. We therefore sought to elucidate the effects of a history of COVID-19 infection and/or vaccination on AT biology as well as serum protein profile in individuals with excessive body mass. To address this, we obtained 27 paired biopsies of VAT and SAT from donors of the Leipzig Obesity BioBank (LOBB). Patients were categorized into three groups based on their serum antibodies to spike and nucleocapsid proteins: group 1 NN (not infected and not vaccinated), group 2 NV (not infected but vaccinated), and group 3 IV (infected and vaccinated).

Recent publications indicate that the SARS-CoV-2 virus can enter adipocytes and AT immune cells by detecting viral RNA in AT [1,2,16,17,18].

Indeed, obesity could contribute to the evolution of COVID-19 infection in different ways, including the promotion of SARS-CoV-2 receptor expression in AT [13]. In some patients, AT may represent a large reservoir for virus replication with increased shedding of virus and inflammatory mediators [14]. Despite our systematic approach, we did not detect SARS-CoV-2 viral RNA in both AT depots of any patient with a previous COVID-19 history. This observation may be explained by an infection date long before bariatric surgery that resulted in the degradation of viral RNA. Importantly, our data are not contradictory to previous work [2] because some studies [13,18] have shown that, regardless of the SARS-CoV-2 infection history, AT does not necessarily have to be positive for viral presence.

Our systematic analysis at the histological and transcriptional levels explored the impact of vaccination-related infection on AT expression, together with serum proteome analysis. Histological adipocyte area analysis indicated that infection itself reduces adipocyte size in both AT depots but is more pronounced in SAT. Interestingly, vaccination did not have an impact on average adipocyte size. Individuals with SARS-CoV-2 infection show an adipocyte size reduction of approximately 40% in SAT and 20% in VAT, further suggesting that adipocytes are targets of SARS-CoV-2 [1,2,10]. A similar picture is also seen in patients with human immunodeficiency virus (HIV) [19]. HIV patients develop a lipoatrophy at all SAT locations (the limbs, trunk, face, and buttocks) [19]. Taken together, viral infection may reduce adipocyte size in SAT and VAT, and vaccination has no effect on the adipocyte size area. Studies on COVID-19 patients have shown dysregulated lipid profiles, which are associated with an increased risk of severe consequences [10,20,21,22]. It has been suggested that modulation of cellular lipid metabolism could be a therapeutic target for SARS-CoV-2 infection [20,22]. However, the influence of the infection on lipid metabolism, lipolysis, and lipogenesis in adipocytes remains unclear.

White AT contributes significantly to inflammation—especially in the context of obesity [23]. In the next step, we focused on transcriptional changes in AT after a COVID-19 infection and/or vaccination. The impact of infection on the differential gene expression in SAT depends on vaccination. Exclusion of vaccination results in a significantly higher number of upregulated genes, indicating an effect of vaccination or the subsequent immune response mechanisms on the gene expression profile in both AT depots. Genes such as *CARMIL2*, *CCR7*, *NIBAN3*, *SAMD3*, and *MS4A1* are consistently present in both comparisons, suggesting that these are the key genes triggered by SARS-CoV-2 infection. Interestingly, all genes are linked to immune-mediated functions.

For example, *CARMIL2*, encoding the essential cytosolic protein for CD28 co-stimulation in T cells, plays a pivotal role in IFN-γ production and viral replication inhibition [24]. Interestingly, CARMIL2 deficiency leads to virus-related skin conditions [25], and three genetic variants and loci are overrepresented in a cohort with very high chronic inflammation markers from Mallorca, emphasizing its connection to diseases predisposing individuals to viral infections and immune dysregulation [26].

Another finding is the elevated expression of *CCR7* in SAT after COVID-19 infection without vaccination. This observation gains significance considering the role of *CCR7* in myofibroblasts in COVID-19 lungs, suggesting a potential response to elevated CC-chemokine ligand 21 (CCL21) signals during prolonged disease phases [27]. High levels of CCL21 are associated with persisting pulmonary impairment post-COVID-19 hospital admission [28], and CCL21 is recognized for its involvement in recruiting CCR7+ T cells to secondary lymphoid organs, orchestrating the adaptive immune response [29]. Notably, studies link obesity to the accumulation of CD11c+ immune cells expressing *CCR7*, contributing to chronic inflammation and insulin resistance [30]. The relationship between *CCR7* and COVID-19 in AT complements existing research focused on blood cells [31]. *CCR7* expression varies in peripheral blood mononuclear cells during different stages of COVID-19, underlining the complexity of the immune responses [32,33]. In leukocytes from whole blood, *CCR7* regulation during COVID-19 exhibits heterogeneous findings [34,35,36,37].

Contrary to *CARMIL2* and *CCR7*, consistent expression of *NIBAN3*, also known as *B-cell novel protein 1* (*BCNP1*), in SAT under consideration of infection with and without vaccination lacks direct associations with obesity, T2D, or COVID-19. It has recently been identified as a new B-cell receptor (BCR) signaling molecule, and BCNP1-deficient mice exhibit impaired B-cell maturation and an enhanced humoral immune response [38]. However, investigations linking NIBAN3 to Parkinson’s disease suggest a potential avenue for further exploration [39,40].

Elevated *MS4A1,* which encodes for CD20, in SAT resulting from infection without vaccination highlights the presence of an immune response. CD20 is activated through microenvironmental interactions by CXCR4/SDF1 (CXCL12) chemokine signaling, and its molecular function has been associated with the signaling capacity of the BCR [41].

The gene *ESRP1*, which is significantly overexpressed in relation to infection and vaccination in SAT, is a crucial component of the spliceosome, and the loss of *ESRP1* leads to female infertility due to oocyte development dysregulation [42]. Although there is no reported connection to COVID-19, altered expression of key spliceosome components, including *ESRP1*, has been mentioned in relation to miR-4454, a microRNA associated with insulin resistance in obesity [43], as well as with caspase-independent cell death [44].

The upregulation in *EOMES*, a transcription factor involved in the T-cell exhaustion process in severe COVID-19 patients, in the comparison describing infected and not vaccinated cases, suggests that *EOMES* may play a role in the development of T-cell exhaustion, as described before [45]. It directly induces IFN-γ and cytotoxicity, enhances IL-10, and antagonizes alternative T-cell fates [46].

Taken together, our transcriptome analysis suggests that COVID-19 affects the inflammation processes in AT (Figure 5). The impact of vaccination on AT inflammation adds a valuable layer to our understanding.

The proximity extension assay (PEA) immunoassay technique was used to measure 269 plasma biomarkers. This was guided by findings that showed significant elevation of markers such as IL-6 and chemokines in COVID-19 intensive care unit (ICU) patients [47].

Indeed, approximately one-third of basal IL-6 serum concentration originates from AT [48]. IL-6 is known to be activated in the context of obesity, and most likely, the SARS-CoV-2 virus could amplify the primed cytokine response in AT [49].

According to these notions, we find lower IL-6 levels in individuals with infection and vaccination. IL-6, promptly and transiently produced in response to infections and tissue injuries, contributes to host defense through the stimulation of acute phase responses, hematopoiesis, and immune reactions. IL-6 promotes specific differentiation of naive CD4+ T cells, thus performing an important function in the linking of innate to acquired immune response [50]. Drugs suppressing IL-6, like tocilizumab, have been shown to have a positive effect on the course of COVID-19 infection [51,52]. Our data suggest that vaccination may have a similar effect, but further in-depth analyses are required to confirm this finding. In particular, we are not able to distinguish which AT cell types may be responsible for the observed group differences in IL-6 serum concentrations and whether tissues other than AT may contribute to higher circulating IL-6 in individuals with a previous COVID-19 infection and vaccination.

Furthermore, our data indicate a lower expression of CA5A, a protein that is a crucial intramitochondrial carbonic anhydrase and catalyzes the reversible conversion of carbon dioxide to bicarbonate/HCO_3_ in mitochondria [53,54]. It provides HCO_3_ for multiple mitochondrial enzymes that catalyze the formation of essential metabolites of intermediary metabolism in the urea and Krebs cycles [54]. This makes it relevant for gluconeogenesis and lipogenesis [55]. It has been shown that CA5A is related to hepatic glucose in T2D and higher levels of fasting glucose, and it therefore has been suggested as a new novel marker for prevalent prediabetes and T2D [56,57]. A CA5A gene mutation has been reported to cause infantile hyperammonemic encephalopathy, which includes hypoglycemia, hyperlactatemia, metabolic acidosis, and respiratory alkalosis. COVID-19 has been frequently associated with blood glucose abnormalities [5,58], which may be reflected by altered serum CA5A expression. CA5A levels are strongly associated with unstable angina and acute myocardial infarction, but without suggesting that it is due to up- or downregulation [59]. This finding is particularly relevant given the increased acute myocardial infarction risk after surviving COVID-19 infection [60].

Previous research has demonstrated that COVID-19 infection can lead to a decrease in adiponectin serum levels [1,12]. Our observations also indicate a similar trend, although it was not statistically significant. The inflammatory response induced by SARS-CoV-2 infection could potentially lead to insulin resistance and disrupt adiponectin signaling, contributing to the Adiponectin Paradox [61].

Our analysis of CA5A, IL-6, and adiponectin serum concentrations may not entirely mirror expression changes in the investigated AT depots. However, adiponectin and IL-6 are both substantially expressed in AT [49,62,63] and closely related to their respective serum concentrations [48]. Both molecules represent inflammatory markers in people with obesity and T2D [62,63,64] and may affect glucose and lipid metabolism [65,66,67,68]. Average adipocyte size and adipocyte hypertrophy are important determinants of altered CA5A and adiponectin expression. However, despite the observed trend of smaller average adipocyte size and AT dysfunction in the infected and vaccinated group, we did not find a significant relationship between adipocyte size and CA5A or adiponectin serum concentrations. We are aware of studies demonstrating that SARS-CoV-2 can replicate in adipocytes and may cause AT inflammation and mRNA expression that subsequently lead to AT dysfunction and altered adiponectin secretion [69]. Single-cell, single-nuclei, and spatial transcriptomics approaches are required in the future to define the contribution of specific AT cell types and adipocyte subclasses on systemic CA5A, IL-6, and adiponectin expression signatures in response to COVID-19 infection. Since CA5A is a more recently described marker for T2D, further investigations are required to decipher a potential mechanistic link between CA5A and AT signature changes.

In summary, a history of COVID-19 infection is associated with reduced average adipocyte size and a distinct AT expression of genes involved in immune response. Particularly, SAT exhibits a gene expression pattern supporting the hypothesis that immune response pathways are activated, underscoring its potential impact on the inflammatory and immune dysregulation observed in individuals with obesity and COVID-19. Further research is needed to unravel the specific mechanisms underpinning these connections and their implications for disease progression and outcomes.

### Limitations

This study has limitations, which are primarily attributed to its retrospective nature. Patient information regarding vaccination status was not obtained. The subjects were selected from a time frame in which vaccination was a requirement for undergoing bariatric surgery, so patients were grouped based on the detection of SARS-CoV-2 antibodies in their blood. The sample size is rather small, with only 27 participants distributed among three groups. On the other hand, the three experimental groups were not different in key parameters, reflecting a similar cardio-metabolic risk profile. In addition, there is an imbalance in gender distribution. With our study design, we are not able to draw conclusions related to a potential interaction of previous COVID-19 infection with obesity with or without T2D or impaired glucose tolerance. Furthermore, it is important to note the lack of information concerning the type of vaccine administered, making it challenging to delineate specific vaccine-induced responses. Moreover, the precise timing of both vaccination and infection is unknown, leading to inconsistency in the progression of immunity and infection in the study participants.

## 4. Materials and Methods

### 4.1. Baseline Characteristics of the Subjects

The cohort consists of 27 metabolically well-characterized individuals of the Leipzig Obesity BioBank (LOBB) who were scheduled to undergo elective cholecystectomy, exploratory laparotomy, or gastric sleeve resection [11]. All patients included in this study had a stable weight, defined as fluctuations of <2% of body weight, for at least 3 months prior to surgery. In addition, clinical and anthropometric parameters were routinely recorded as described previously [70]. According to American Diabetes Association (ADA) criteria [71], 37.0% of the patients were diagnosed with T2D, 44.4% had normal glucose tolerance (NGT), and 18.5% exhibited a prediabetes state such as impaired glucose tolerance (IGT). Moreover, patients diagnosed with acute or chronic hepatic, inflammatory, infectious, or neoplastic diseases were excluded from this study.

The cohort was classified into distinct groups based on their SARS-CoV-2 antibody detection. A negative detection for both N- and S-proteins signifies an individual who is neither vaccinated nor infected (NN, controls, group 1). Positive detection for S-protein but negative for N-protein indicates that the individual has been vaccinated but is not infected (vaccination, group 2, NV). On the other hand, a positive detection for both S- and N-protein represents an individual who has been both vaccinated and infected (group 3, vaccination and infection, V). We aimed to ensure comparability between groups by carefully matching participants based on their BMI and age, with the goal of achieving similar means and standard deviations within each group. Furthermore, we aimed to balance the distribution of sexes across groups to minimize potential confounding effects. However, achieving equal representation of sexes in Group 3 IV proved challenging due to the limited number of participants in this particular group.

The study was approved by the ethics committee of the University of Leipzig (Approval numbers: 159-12-21052012 and 017-12-23012012). The study design follows the Declaration of Helsinki, and all subjects in this study gave written informed consent to participate in medical research.

### 4.2. SARS-CoV-2 Antibody Detection

This study categorized the samples into different groups based on their SARS-CoV-2 antibody profiles as detected by the Institute of Laboratory Medicine, Clinical Chemistry and Molecular Diagnostics at University Leipzig, Leipzig, Germany. Antibodies against the spike protein indicated prior vaccination, while the presence of antibodies against both the spike and nucleocapsid proteins suggested a combination of vaccination and infection.

During the study period, several COVID-19 vaccines were available in Germany. These included Comirnaty (BNT162b2), Spikevax (mRNA-1273), Jcovden (Ad26.COV2.S), and Vaxzevria (ChAdOx1-S). Comirnaty and Spikevax are mRNA vaccines, while Jcovden and Vaxzevria are vector-based vaccines. All these vaccines target the spike protein of the SARS-CoV-2 virus to stimulate an immune response [72].

Comirnaty received EU approval on 21 December 2020, followed by Spikevax on 6 January 2021. Jcovden received EU approval on 11 March 2021, and Vaxzevria received approval on 29 January 2021 [72]. It is important to note that this study classified the samples based on their SARS-CoV-2 antibody profiles, specifically the presence of antibodies against the spike and nucleocapsid proteins.

### 4.3. Viral RNA Detection in Adipose Tissue

Viral RNA was extracted from frozen SAT and VAT using the previously described method [2]. Homogenization was performed in a Precelly homogenizer (Precellys 24 tissue homogenizer, Bertin Technologies SAS, Montigny-le-Bretonneux, France) with 1 mL of sterile PBS, following the manufacturer’s instructions. Subsequently, viral RNA was isolated using the QIAamp Viral RNA Mini Kit (QIAGEN GmbH, Hilden, Germany), according to the manufacturer’s protocol. Detection of lineage B beta coronavirus (lineage B-βCoV) and SARS-CoV-2 was conducted using qRT-PCR with the RealStarSARS-CoV-2 RT-PCR Kit 1.0 (altona Diagnostics GmbH, Hamburg, Germany), following the manufacturer’s instructions.

### 4.4. Characterization of Adipose Tissue Samples

During surgery, AT samples were obtained in pairs from the abdominal omental (visceral, VAT) and subcutaneous (SAT) fat depots from 27 Caucasian participants, men (n = 12) and women (n = 15), during elective laparoscopic bariatric surgery as described previously [73]. Whole AT was immediately frozen in liquid nitrogen after explanation and stored at −80 °C. Histological analyses were performed according to previously described methods [70]. Briefly, AT samples were fixed in 4% formaldehyde at room temperature and embedded in paraffin.

Sections were stained with hematoxylin/eosin and analyzed using a blinded approach with a Keyence BZ-X800 microscope and Keyence BZ-X800 Analyze 1.1.1.8r (KEYENCE DEUTSCHLAND GmbH, Neu-Isenburg, Germany) software to identify adipocytes and measure their size and perimeter in multiple sections.

### 4.5. RNA-Sequencing of Adipose Tissue

RNA extraction for transcriptomic analysis from AT was performed using the RNeasy Lipid Tissue Mini Kit (QIAGEN GmbH, Hilden, Germany) according to manufacturer’s instructions.

RNA samples were quantified, and their integrity was checked before performing rRNA and globin depletion, followed by library preparation and sequencing using Illumina NovaSeq 6000 by the Leipzig laboratory of GENEWIZ Germany GmbH. RNA library preparation was carried out using the NEBNext Ultra II Directional RNA Library Prep Kit for Illumina following the manufacturer’s instructions (NEB, Ipswich, MA, USA). The samples were sequenced using a 2 × 150 Pair-End (PE) configuration.

Raw sequencing reads were subjected to data preprocessing to ensure high-quality data for downstream analysis. The tool Trimmomatic [74] was utilized to trim adapter sequences and remove low-quality nucleotides. Only reads with a minimum quality threshold of 30 and a minimum length of 36 nucleotides were retained for further analysis. The trimmed reads were aligned to the human reference genome using the STAR aligner v.2.5.2b [75]. To account for potential multi-mapping issues, we allowed up to 100 multiple alignments per read. To ensure the quality of the alignment, standard pre- and post-mapping quality control steps were conducted using FASTQC v.0.11.4 [76]. Gene expression levels were quantified by assigning mapped reads to each genomic feature using FeatureCounts v2.0.1 [77]. In cases where reads were mapped to multiple locations, a fractional counting method was employed to accurately account for multi-mapped reads. Prior to differential expression analysis, data normalization was performed to remove technical biases and ensure comparability across samples. The R package DESeq2 v3.9 [78] was employed for normalization and differential expression analysis. This widely recognized package is known for its robust performance and ability to handle sequencing data with statistical rigor. To address potential batch effects and other unwanted variations that may confound the results, the sva v3.48.0 R package [79] was utilized. We calculated the first five surrogate variables and incorporated them into the linear model during the differential gene expression analysis. Shrinkage estimation for dispersions and fold changes to improve stability and interpretability of estimates were applied using the apeglm R package v1.22.1 [80]. Significant changes in gene expression were assumed for an adjusted *p*-value (padj)  <  0.05 and a log2 fold change (FC) > |0.5|. The R package Clusterprofiler [81] was used to perform GO and KEGG enrichment analyses.

### 4.6. Serum Proteomics Profiling

The selected frozen plasma samples were aliquoted and sent to Olink^®^ Proteomics AB for proteomic analysis using the proximity extension assay technology for three different panels. The Olink^®^ Inflammation, Cardiometabolic, and Cardiovascular II Panels were utilized to measure the levels of 276 proteins (Appendix A). The resulting data were provided as Normalized Protein Expression (NPX) values. Differential Expression (DE) analysis was conducted using the R package OlinkAnalyze v. 3.5.1 provided by Olink (Olink Proteomics AB, Uppsala, Sweden).

### 4.7. Serum Adiponectin Measurement

Adiponectin serum concentrations were quantified in duplicates utilizing the Adiponectin (human) Enzyme-Linked Immunosorbent Assay (ELISA) Kit procured from Adipogen (Adipogen AG, Füllinsdorf, Switzerland) according to the manufacturer’s instructions. The Assay sensitivity was <0.5 ng/mL, and the inter-assay and intra-assay coefficients of variation were less than 3.8% and 5.5%, respectively.

### 4.8. Statistical Analyses

The data are presented as mean values accompanied by the standard error of the mean (±SEM) unless stated otherwise. Normality of the data distributions was evaluated using the Shapiro–Wilk test before any inferential analyses were conducted. A *p*-value greater than 0.05 in this test indicated that the data were normally distributed. For variables that deviated from normality, logarithmic transformation was utilized unless otherwise indicated.

To determine any potential differences among groups, we conducted comparative statistical analyses using analysis of variance (ANOVA) for normally distributed data. We then applied the Tukey method for post hoc testing for multiple comparisons. If non-normally distributed data were present, Kruskal Test with Dunn’s post hoc analysis was employed. Based on *p*-values adjusted for significance levels, we set the threshold for statistical significance at less than 0.05. This comprehensive approach to statistical analysis ensures rigorous examination of the data, considering both distributional characteristics and appropriate adjustments for multiple comparisons, thus enhancing the reliability and validity of our findings.

## 5. Conclusions

The impact of COVID-19 infection on AT, especially in SAT, is significant, with more pronounced effects observed in unvaccinated individuals with a history of COVID-19. Enrichment analysis suggests activation of immune system-related pathways in both AT depots. In individuals with obesity and T2D, the inflammation observed by gene expression in AT during infection suggests a potential link between obesity-related inflammation and the immune response to COVID-19. We postulate that the effects of COVID-19 on AT may contribute to the increased severity and adverse outcomes of COVID-19 in people with obesity. Additionally, infection-induced adipocyte shrinkage may exacerbate metabolic dysfunctions, potentially worsening outcomes in these populations. The impact of COVID-19 on individuals with obesity and T2D highlights the need for targeted interventions to mitigate its effects.

## Figures and Tables

**Figure 1 ijms-25-02908-f001:**
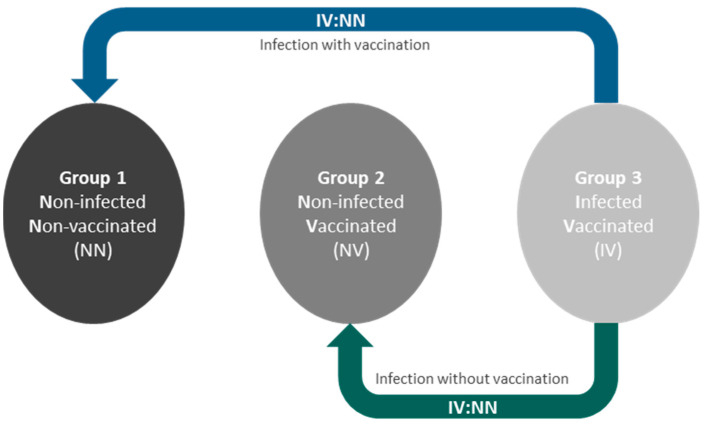
Study design and presentation of the group comparisons. To isolate infection-specific effects, we compared group 3 (infected and vaccinated, IV) with group 2 (vaccinated but not infected, NV). Additionally, we assessed the effects of vaccination in the context of infection by comparing group 3 with group 1 (neither infected nor vaccinated, NN).

**Figure 2 ijms-25-02908-f002:**
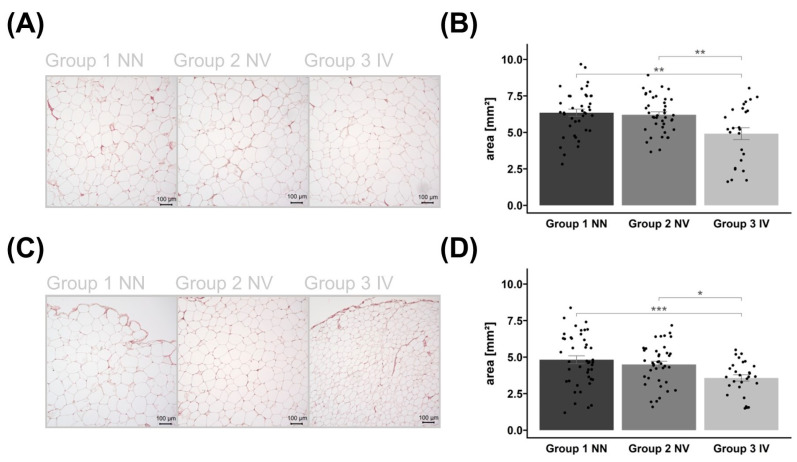
Histological examination of the adipocytes. Representative examples of HE-stained subcutaneous (SAT) (**A**) and visceral adipose tissue (VAT) (**C**) images of control group 1, non-infected and non-vaccinated (NN); group 2, vaccinated but non-infected (NV); as well as group 3, infected and vaccinated (IV). The corresponding bar plots of adipocyte areas are presented as mean ± SEM with individual measurements as dots (**B**/**D**). The same applies to VAT in (**C**,**D**). Images are in 10× magnification, captured using Keyence BZ-X800 microscope (KEYENCE DEUTSCHLAND GmbH, Neu-Isenburg, Germany), and measured by Keyence BZ-X800 Analyzer (KEYENCE DEUTSCHLAND GmbH, Neu-Isenburg, Germany). Significant *p*-values are shown as * *p* < 0.05, ** *p* < 0.01, and *** *p* < 0.001.

**Figure 3 ijms-25-02908-f003:**
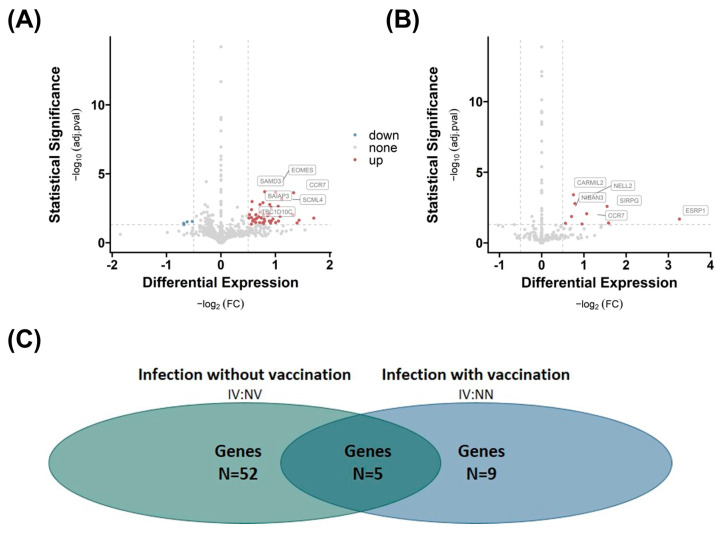
Effects of COVID-19 infection or vaccination on differential gene expression in subcutaneous adipose tissue (SAT). Differential gene expression of group 3 IV versus group 2 NV (**A**) and group 1 NN (**B**) in SAT. Differentially expressed genes are determined by comparing absolute log2 fold changes greater than |0.5|, and statistical significance is indicated by adjusted *p*-values (adj.pval) less than 0.05. The number of significant differentially expressed genes in both comparisons (N (**A**) = 52, N (**B**) = 9), as well as the number of overlapping genes (n = 5) between the two comparisons, is shown in (**C**).

**Figure 4 ijms-25-02908-f004:**
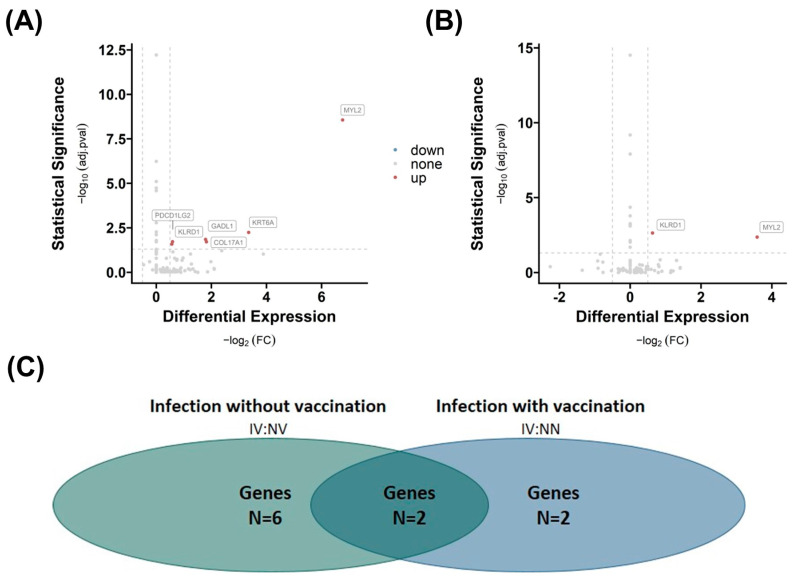
Effects of infection and vaccination on differential gene expression in visceral adipose tissue (VAT). Differential gene expression of group 3 IV versus group 2 NV (**A**) and group 1 NN (**B**) in VAT. Differentially expressed genes are determined by comparing absolute log2 fold changes greater than |0.5|, and statistical significance is indicated by adjusted *p*-values (adj.pval) less than 0.05 using the prescribed methodology. The number of significant differentially expressed genes in both comparisons (N (**A**) = 6, N (**B**) = 2), as well as the number of overlapping genes (n = 2) between the two comparisons, is shown in (**C**).

**Figure 5 ijms-25-02908-f005:**
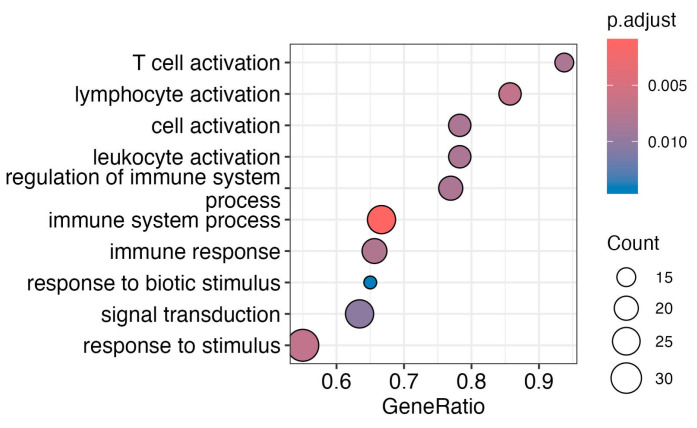
Enrichment analysis from transcriptomic effects of infection without vaccination in SAT. Pathway analysis of differentially expressed genes analysis of group 3 versus group 2 reveals effects on immune cell response, such as T-cell and lymphocyte activation, as well as broader cell and leukocyte activation. Additionally, pathways like the regulation of immune system processes, including immune system processes and immune response, are significant. Active cellular signaling is demonstrated by biotic stimulus, signal transduction, and response to stimulus pathways. Only differentially expressed genes with an adj. *p*-value < 0.05 were used for gene enrichment analysis based on the Gene Ontology (GO) database of biological processes.

**Figure 6 ijms-25-02908-f006:**
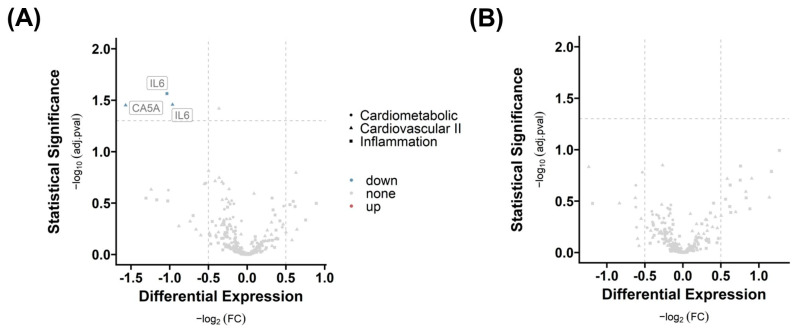
Differential serum protein expression in response to vaccination and infection. Differential proteins with respect to (**A**) infection without vaccination (group 3 versus group 2) and (**B**) infection including vaccination (group 3 versus group 1) are presented. The graphic shows the targeted analysis of 276 proteins in serum across three distinct panels, classified by shape. For the identification of differentially expressed serum proteins, we implemented a log2 fold change (FC) cut-off of greater than |0.5|, and significance was determined by an adjusted *p*-value (adj.pval) of less than 0.05.

**Figure 7 ijms-25-02908-f007:**
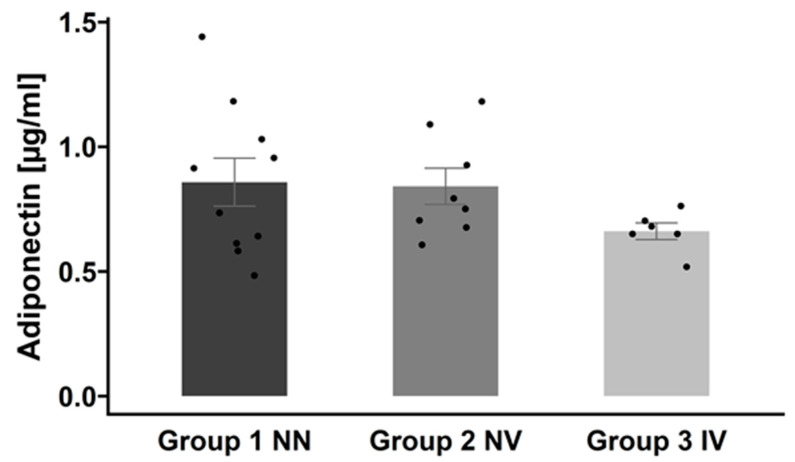
Adiponectin serum concentrations across the three groups. Adiponectin levels are presented as mean ± SEM. Group 1 NN: non-infected and non-vaccinated; group 2 NV: non-infected and vaccinated; group 3 IV: infected and vaccinated.

**Table 1 ijms-25-02908-t001:** Anthropometric and metabolic characteristics of the study cohort.

	Group 1 Controls (NN)	Group 2 Vaccination (NV)	Group 3 Infection/Vaccination (IV)
	n = 10	n = 10	n = 7
Men/women	5/5	5/5	2/5
Age [years]	42.8 ± 10.4	43.0 ± 9.5	42.7 ± 8.9
Body weight [kg]	138.5 ± 34.3	141.8 ± 20.3	127.0 ± 21.9
Height [m]	1.7 ± 0.1	1.7 ± 0.1	1.7 ± 0.0
BMI [kg/m^2^]	44.1 ± 7.1	44.3 ± 4.9	43.8 ± 5.8
Body fat mass [%]	48.6 ± 9.6	46.9 ± 8.1	50.3 ± 7.8
Creatinine [μmol/L]	73.1 ± 17.6	76.3 ± 12.8	75.1 ± 10.5
C-reactive protein [mg/L]	9.5 ± 10.2	6.6 ± 7.1	4.0 ± 1.5
Albumin [g/dl]	4.5 ± 0.3	4.6 ± 0.2	4.5 ± 0.2
FPG [mmol/L]	6.6 ± 3.9	6.2 ± 1.6	4.7 ± 0.4
HOMA-IR Index	9.6 ± 9.8	4.8 ± 5.6	1.8 ± 0.9
HbA_1c_ [%]	6.1 ± 1.4	6.4 ± 1.2	5.5 ± 0.5
Total cholesterol [mmol/L]	4.6 ± 1.3	4.3 ± 1.3	4.3 ± 0.6
HDL cholesterol [mmol/L]	1.5 ± 0.6	1.2 ± 0.2	1.3 ± 0.3
LDL cholesterol [mmol/L]	2.8 ± 1.1	2.8 ± 1.2	2.6 ± 0.7
Triglycerides [mmol/L]	1.5 ± 1.5	1.6 ± 0.5	1.4 ± 0.9
Diabetes status			
Normal glucose tolerance [%]	30	40	71
Type 2 diabetes [%]	40	50	14
Impaired glucose tolerance [%]	30	10	14

Data are shown as mean ± SD per group. Statistical differences between groups were analyzed using ANOVA-Tukey for normally distributed parameters, indicated by a mean *p*-value per measurement tested with Shapiro–Wilk. The analysis has shown no statistical differences. NN: non-infected and non-vaccinated; NV: non-infected and vaccinated; IV: infected and vaccinated; BMI: body mass index; FPG: fasting plasma glucose; HOMA-IR: homeostasis model assessment–insulin resistance; HbA_1c_: glycated hemoglobin; HDL: high-density lipoprotein; LDL: low-density lipoprotein.

## Data Availability

All data from this study are contained within the published article and its Appendix A. The RNA-seq data can be accessed at NCBI’s Sequence Read Archive [82] under BioProject ID PRJNA1079046 (https://www.ncbi.nlm.nih.gov/bioproject/PRJNA1079046).

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
