# Peer review of "Consequences of COVID-19 on Adipose Tissue Signatures"

_ijms, 2024, doi:10.3390/ijms25052908_

Round 1

Reviewer 1 Report

Comments and Suggestions for Authors

Review for ijms-2876676

Title: Deciphering the Role of Adipose Tissue in COVID-19 Pathogenesis

Authors: Sontje Krupka , Anne Hoffmann , Mariami Jasaszwili , Arne Dietrich, Esther Guiu-Jurado , Nora Klöting ,  Matthias Blüher

A study by Krupka et al. describes the effects of  SARS-CoV-2 infection and/or vaccination on adipose tissue biology in obese individuals. I think that the title of the paper should be changed in this light. The idea of this study is interesting and worthy of attention because the mechanisms underlying the link between obesity and complications upon SARS-CoV-2 infection are not clear enough.  The article provides some new information which was not mentioned by other similar articles.  Although the manuscript is generally correct several issues need to be resolved before the article is ready for publication.

Several studies have addressed the topic of this manuscript and have indicated that obesity could contribute to the evolution of infection in multiple ways including the promotion of enhancing expression of SARS-CoV-2 receptors in adipose tissue [Basolo, A., Poma, A.M., Bonuccelli, D. et al. Adipose tissue in COVID-19: detection of SARS-CoV-2 in adipocytes and activation of the interferon-alpha response. J Endocrinol Invest 45, 1021–1029 (2022) ] or, in certain patients, by representing a large reservoir for virus replication with increased shedding of virus and inflammatory mediators,  [Ryan PM, Caplice NM (2020) Is Adipose tissue a reservoir for viral spread, immune activation, and cytokine amplification in coronavirus disease 2019? Obesity (Silver Spring) 28(7):1191–1194]. Studies of Basolo et al. and Moser et al. (Moser J, Emous M, Heeringa P, Rodenhuis-Zybert IA. Mechanisms and pathophysiology of SARS-CoV-2 infection of the adipose tissue. Trends Endocrinol Metab. 2023 Nov;34(11):735-748.) have shown that, regardless of the general SARS-CoV-2 infection, adipose tissue does not necessarily have to be positive for viral presence. Therefore, I am asking the authors are they are sure that the tested samples of adipose tissue from COVID-19 patients are all positive or all negative for the virus? Are infected adipose tissue with SARS-CoV-2 and its cellular responses different compared to non-virus-infected adipose tissue? This should be explained.

The study was carried out on a small number of samples, 27 of them, which were additionally divided into three groups within which both sexes were present. The study lacks a group of obese patients who were infected and not vaccinated as a confirmation of the results and derived conclusions obtained with the IV group. In the Material and Methods section, the authors state that 37% of the respondents had DM2. How they are distributed by group, whether there were patients with DM2 within each group, and whether there are differences in the examined parameters in individuals with and without DM2? What does it mean in subsection 4.1, which refers to baseline characteristics of subjects: Error! Reference source not found? I suggest that Table 1 be corrected in the paper according to the given remark. Or, to show in the supplement the results of the tested parameters which show that there is no influence of DM2 and therefore the summary results could be shown as in Table 1.

Obesity induces adipose tissue expansion via increased adipocyte number or size, and besides its storing capacity, WAT possesses important endocrine functions which are represented by the release of different factors including cytokines such as IL-6 (one-third of basal concentration in human circulation comes from WAT, especially visceral WAT) (Wueest S, Konrad D. The controversial role of IL-6 in adipose tissue on obesity-induced dysregulation of glucose metabolism. Am J Physiol Endocrinol Metab. 2020 Sep 1;319(3):E607-E613).. IL-6 is known to be preactivated in the context of obesity and it seems that virus could amplify the already primed cytokine response in adipose tissue (Jonas, M.I.; Kurylowicz, A.; Bartoszewicz, Z.; Lisik, W.; Jonas, M.; Wierzbicki, Z.; Chmura, A.; Pruszczyk, P.; Puzianowska-Kuznicka, M. Interleukins 6 and 15 Levels Are Higher in Subcutaneous Adipose Tissue, but Obesity Is Associated with Their Increased Content in Visceral Fat Depots. Int. J. Mol. Sci. 2015, 16, 25817-25830). Figure 6 shows that infected patients when vaccinated have a lower level of serum IL-6, which is expected. But how the authors relate it to adipose tissue, given that IL-6 is produced and secreted by different cells and tissues? Also, how is the decrease in the serum levels of CA5A and adiponectin in the infected and vaccinated group related to a decreasing trend in adipocyte size and probably AT dysfunction? Studies evidenced that SARS-CoV-2 can replicate into the adipocytes and cause AT inflammation, which alters mRNA expression in these cells and leads to AT dysfunction with modified adiponectin secretion (Barbalho, S.M.; Minniti, G.; Miola, V.F.B.; Haber, J.F.d.S.; Bueno, P.C.d.S.; de Argollo Haber, L.S.; Girio, R.S.J.; Detregiachi, C.R.P.; Dall’Antonia, C.T.; Rodrigues, V.D.; et al. Organokines in COVID-19: A Systematic Review. Cells 2023, 12, 1349). Please, explain this in discussion section. Also, is the labeled position of IL-6 and CA5A missing from Figure 6A?

Minor point:

1.      Page 10, line 308: COVID-19 instead Covid-19

2.      Page 13, line 441: Check reference . Bolger et al (53) instead Trimmomatic (53)

Reviewer 2 Report

Comments and Suggestions for Authors

Journal: IJMS    Manuscript ID: ijms-2876676

Authors: Sontje Krupka et al.

Title: "Consequences of COVID-19 on Adipose Tissue Signatures"

1.     The authors should clarify the three groups of participants in the abstract. Moreover, providing more specific information regarding the main findings would enhance its informativeness to the reader.

2.     Do the authors have data from patients with COVID-19 infection who were not previously vaccinated?

3.     Could the authors clarify how the matching was performed, including whether a specific approach was applied among the groups? Additionally, what do the authors mean by "if possible" in the sentence "We matched groups by age, bodyweight, and sex if possible"? Furthermore, did the authors match based on weight or BMI?

4.     When applicable, authors should include in the figures and tables the overall p-value corresponding to the ANOVA/Kruskal test for the relevant analyses (which also justify post-hoc analyses).

5.     It would be informative if the authors could add the diseases/disorders of the patients in Table 1 (e.g., T2D, prediabetes, hypertension). Additionally, in the footnote, the authors mention that they have performed analyses among the groups. If no significant difference was observed, it should be briefly noted in the table; otherwise, they should annotate them accordingly.

6.     Please ensure that all the references are correctly cited: "(Error! Reference source not found.)"

Round 2

Reviewer 1 Report

Comments and Suggestions for Authors

In the present form, the manuscript has been sufficiently improved to warrant publication in IJMS.

Reviewer 2 Report

Comments and Suggestions for Authors

Journal: IJMS    Manuscript ID: ijms-2876676 (Revised)

Authors: Sontje Krupka et al.

Title: "Consequences of COVID-19 on Adipose Tissue Signatures"

The authors of the present article have satisfactorily responded to my comments and suggestions and made the necessary changes to the paper. There are no further comments.